# 'Scrupulous and Timid Conformism': Ireland and the Reception of the Liturgical Changes of Vatican II

**Gary Carville**

Mater Dei Centre for Catholic Education, Dublin City University, 9 Dublin, Ireland; gary.carville@dcu.ie

**Abstract:** The Second Vatican Council and, in particular, its Constitution on the Sacred Liturgy, changed much in the daily life of the Church. In Ireland, a country steeped in the Catholic tradition but largely peripheral to the theological debates that shaped Vatican II, the changes to liturgy and devotional practice were implemented dutifully over a relatively short time span and without significant upset. But did the hierarchical manner of their reception, like that of the Council itself, mean that Irish Catholics did not receive the changes in a way that deepened their spirituality? And was the popular religious memory of the people lost through a neglect of liturgical piety and its place in the interior life, alongside what the Council sought to achieve? In this essay, Dr Gary Carville will examine the background to the liturgical changes at Vatican II, the contribution to their formulation and implementation by leaders of the Church in Ireland, the experiences of Irish Catholic communities in the reception process, and the ongoing need for a liturgical formation that brings theology, memory, and practice into greater dialogue.

**Keywords:** Vatican II; liturgy; popular piety; Catholic Church; Ireland; vernacular; reform





## 1. Introduction

The Constitution on the Liturgy, *Sacrosanctum Concilium*, was the first document promulgated by Vatican II, on 4 December 1963. Pope Paul VI declared that its primary place in the council gave expression to 'the recognition of a right order of values and duties: God in the first place, prayer our first duty, the liturgy the first school of spirituality' (Hebblethwaite 1994). The Constitution on the Liturgy was 'with little doubt the most decisive for most Catholics' in the daily life of the church (Hastings 1999). It was a document that led to the first visible and audible signs of change following the council. In many respects, *Sacrosanctum Concilium* remains the iconic document of Vatican II.

Liturgy concerns public worship. It 'designates the official public worship of the church' (Collins 1987). By official, it is meant that it is authorised by and celebrated in communion with the local bishop, according to norms approved by the Holy See. By public, it is meant that it is celebrated as an activity by a visible assembly of believers. The word liturgy is derived from two Greek words—*laos* (people) and *ergon* (work). Christian usage of the term narrowed its meaning down to the public worship of the church. The liturgy is identifiable by the rites celebrated and by the liturgical books. These are centered on the eucharistic mystery. In the Roman Catholic Church, the eucharist is the memorial of the life, death, and resurrection of Christ, through whom, according to Catholic belief, salvation is found. From the earliest times, therefore, the Church has celebrated this memorial as a mystery, known as the paschal mystery, because it was brought to completion at the celebration of the Passover, the first Easter. This shift in emphasis was to feature strongly in the post-Vatican II reforms.

Rituals, such as a celebration of the Liturgy of the Word or the Prayer of the Church or devotions such as the rosary or benediction, 'are either an anticipation or an extension of the eucharistic assembly' (Collins 1987). In the Irish context, this is an important consideration in terms of the reception of the liturgical changes and their integration with the memory of

the people. What was that memory in Ireland, and, importantly, what is that memory in the aftermath of the council? How did the emphasis on personal and communal participation in the liturgical rites integrate with the memory of a people formed by the emphasis on the personal journey to God, in the tradition of devotional practices of popular piety, and liturgies that were celebrated almost exclusively in a language incomprehensible to the great majority of them?

## 2. Liturgical Life in Ireland Pre-Vatican II

In 1929, Irish Catholics celebrated the centenary of Catholic Emancipation with a series of religious events in Dublin. These were but a dress-rehearsal for what was to come. Dublin was the location for the 31st International Eucharistic Congress in June 1932, a major event not just for Dublin but also for the newly independent Irish State as it provided an opportunity not just for spectacle but also for an expression of national and religious pride. It also demonstrated to the world the new State was capable of undertaking such a major event. Moreover, the Congress was notable for the highly visible levels of popular piety and devotion. A Eucharistic Congress League organised a 'Crusade of Prayer' for the success of the congress, and a triduum was held in churches across Dublin, while midnight Masses were celebrated in many churches across the country on the night before the official opening. An estimated one million people attended the Pontifical High Mass in the Phoenix Park, which was followed by a magnificent Eucharistic Procession along the quays and streets to O'Connell Bridge where the Papal Legate imparted Benediction of the Blessed Sacrament. Film reel footage shows the vastness of the crowds. Dublin and Ireland were on show to the world (O'Dwyer 2007). Less known were the 'active participation' moments around the country such as when people in Castleblayney knelt on the streets of the town during the Mass in the Phoenix Park 'as devoutly as if they were in sight of the altar', thus recalling a practice of spiritual communion that dated from the Penal Period (Carville 2011).[1]

For the people of Dublin, attendance at and fulfilment of devotional requirements was part of life and memory. The levels of devotional participation may have had their roots in the upsurge of Catholic life following the Irish Famine of the 1840s and the subsequent Synod of Thurles in 1850 (Larkin 1972; Connolly 1985). However, by the second quarter of the twentieth century, such practices were the rhythm of Irish society. Writing of Dublin's Catholic life as experienced by laity in the 1930s, Maurice Hartigan paints a picture of religious zeal (Hartigan 2000). This fervour was reflected not only in attendances at Mass and other devotions but also in the work of sodalities and confraternities that sought to consolidate the effect of parish missions through regular attendance by the laity at the various sacraments. Devotions popular among the laity included those to the Sacred Heart, Marian Devotions such as the praying of the rosary and pilgrimages to Marian shrines, devotion to St Brigid and also to St Anne, and the growing cults of Matt Talbot, Fr Charles of Mount Argus, Fr John Sullivan, and Fr Willie Doyle. A devotion of particular interest, and one which was honoured in Dublin and across Ireland, was that of the blessing of throats on 3 February, the feast of St Blaise. All of these devotions were aided by a large growth in the availability of devotional printed material such as the *Irish Messenger of the Sacred Heart* (founded in 1888), *Virgo Potens* (to promote the Miraculous Medal Crusade), and various works on the lives of the saints and religious figures. Hartigan points in particular to the practice of Catholics making written promises, published in the *Irish Messenger of the Sacred Heart* to perform certain acts as thanksgiving for favours received. In a one-year period (1927–28), some 20% of promises given were to make visits to the Blessed Sacrament, 15% to keep one hour of silence, 12% to attend weekday Mass, 12% to pray the rosary, and 11% to make an Act of Spiritual Communion, while only 6% promised to go to receive Holy Communion. Therefore, prayer in the presence of the Blessed Sacrament was seen as being more spiritually satisfying than receiving the Eucharist itself. For many, receiving Holy Communion involved preparation—mainly Confessions and observation of the fasting regulations—and this led to a minimalist approach. A study of the border town

of Castleblayney in the 1950s shows that a minimalist approach had developed over many decades, with the main emphasis being on the fulfillment of the attendance obligation. Indeed, it cites examples of Holy Communion being distributed before Mass began and also of cases where the distribution of Holy Communion was omitted altogether (Carville 2011). All of this was despite the efforts of more recent Popes such as Pius X (1903–1914) to encourage frequent reception of the Eucharist. Catholics in Ireland were reticent to receive the Blessed Sacrament too often or casually. Catholics in Ireland preferred to live their faith through prayer and devotion rather than acts of charity. For example, the same survey of promises shows that only 2% offered to visit the poor and only 1% offered to work for the poor (Hartigan 2000). People did not form a connection between the Eucharist and the mission at its heart: to love and serve God and one another. This connection was emphasised in catechesis ever since the medieval period, where the celebration and explanation of the Word of God, the sacrifice of the Body and Blood of Christ, and concern for the poor were central. The eleventh-century *Leabhar Breac* has a homily that states that if we receive the Body and Blood of Christ 'with faith and perfect works, with alms and mercy for the Lord's poor and destitute, that sacrifice will make us holy . . . ' (McNamara 2012).

Writing in the 1950s, the French diplomat Jean Blanchard noted that the 'religious attitude of the Irishman is explained . . . by the country's history', adding that the church 'remains a Church of the ordinary people, not of any privileged class'. (Blanchard 1963). Any look at the years before or after 1948 will show that the picture of religious fervour painted by Hartigan of the 1920s and 1930s still remained vibrant. The Holy Year of 1950 and the Marian Year of 1954 were highpoints for the demonstration of personal and public piety. Even public bodies and trade unions played public parts in these, such was the culture of the country at the time. Among the manifestations of this was the erection of crosses in public spaces. In 1950, Drogheda Corporation, a local authority, erected a forty-feet-high cross on the south side of the river Boyne, which was illuminated at night and could be seen for miles. Similarly, in Co. Kerry, a cross was erected on top of Ireland's highest mountain, Carrantuohill, the unveiling of which was attended by 1500 people from all over Kerry (Fuller 2002, p. 24). In October 1953, the Irish bishops urged people to erect statues in honour of the Blessed Virgin Mary during the forthcoming Marian Year. This was willingly taken up, with statues and many grottoes of Our Lady of Lourdes being erected in virtually every parish in the country. Workplaces too became a location for the erection of statues, including at Dublin Port and other prominent places. The staff of the *Irish Independent* newspaper erected a wooden statue of Our Lady, known as Our Lady of Dublin, in a prominent place at their head office, and the rosary was prayed by staff at midnight each night.

Pilgrimages too became a greater feature of the religious experience of Irish people in the 1950s. The first ever national pilgrimage to Rome was held in April–May 1950, led by the Archbishop of Armagh, John D'Alton. It numbered 1000 people and was followed by a second pilgrimage of over 2000 people, led by Archbishop John Charles McQuaid of Dublin, in October. This second pilgrimage included a delegation from the Workers Union of Ireland, and the union presented Pope Pius XII with a chasuble on behalf of all the workers of Ireland. Over 8000 people from all over Ireland took part in a pilgrimage to Knock in August 1950. Local newspapers in the west reported that an estimated 100,000 people took part in the Holy Year pilgrimage to Croagh Patrick in Co Mayo on 30 July 1950, to be followed by 75,000 in 1951—a figure which was maintained into the 1960s (Fuller 2002). The Holy Year of 1950 also saw the national shrine of St Patrick's Purgatory at Lough Derg, Co. Donegal, welcome 30,963 pilgrims for its three-day pilgrimage exercises in 1950 and 34,645 in 1952, its highest ever (McGuinness 2000).

Outside of attendance at Mass, the greatest example of Catholic devotion in homes and in public settings was the recitation of the rosary. Its place in the daily routine of people and homes was central. John Healy reflects that when it came to rosary time:

> There was no need for words now. The clock would strike ten. Grandma would put her sewing aside. From a nail on the wall she'd take the big Rosary beads

> . . . we got up from our seats, knelt down with our backs to the fire and one another, leaned our elbows on the seat . . . and made the responses. It was so in my mother's day and it would be so in my childhood days. (Healy 1978)

In April 1954, a Family Rosary Crusade was launched by the Archbishop of Tuam, Patrick Walsh, and throughout the following month of May, a traditional month of Marian devotion, the Irish–American priest Fr Patrick Peyton travelled around Ireland holding rallies in various places attended by tens of thousands (Fuller 2002; Irish Catholic Directory 1955).

Another feature of devotional practice in Ireland at this time was the procession in towns and villages on the feast of Corpus Christi, a feastday introduced by Pope Urban IV in 1264 and promoted by St Juliana of Liège, a Norbertine mystic of the period. The feast, devoted to the celebration of the Body and Blood of Christ, became universal quite quickly, and there are references to elaborate processions on the feast in medieval Ireland, with the Mayor of Dublin Thomas Collier issuing detailed guidelines for the participation of the city's guilds in 1498. Some decades earlier, 'the corporate, multifaceted nature of the feast' is evident from its celebration in Drogheda in 1412 when a local Dominican friar used the occasion to bring together two rival factions in the town, thus ending a division that had led to bloodshed (Ó Clabaigh 2012).

Over the decades, Corpus Christi processions were significant community events; people tidied and decorated their homes along the route with religious pictures and altars, flags, buntings, and banners were erected; the children—particularly the First Holy Communion children—had a special part to play in the procession, as did the members of the Defence Forces and Gardaí. At the end of each procession, Benediction of the Most Blessed Sacrament was imparted. In Ulster, by the late 1940s, Catholics were becoming confident enough to have public celebrations of the feast. In 1948, for example, Bishop Eugene O'Callaghan of Clogher led the procession in Monaghan town, by then considered a Catholic stronghold (Livingstone 1980). The first Corpus Christi procession in the neighbouring border town of Castleblayney was held in 1952, where a number of homes flew the Eucharistic Congress flag of twenty years earlier (Carville 2011, pp. 150–51). The Corpus Christi procession in Dublin in 1954 was notable, with over 25,000 people taking part (Irish Catholic Directory 1955). An account of the Corpus Christi processions during the 1960s in Dingle, Co Kerry tells how the social and hierarchical make-up of Irish life was reflected:

> Men and women were segregated. The women's group was sub-divided. First came the Legion of Mary. Each member wore a blue sash in honour of the Blessed Mother. The pioneer group . . . followed them. Then came the other ladies and young children. The girls from Coláiste Íde were allowed out from the residential college in Burnham for the occasion. The men's group were similarly divided in social and religious ranks. The Legion of Mary, the St Vincent de Paul Society, Pioneers, teachers, professions, farmers and finally the fishermen. (O'Flaherty 2016)

Alongside religious and social change, the decline, though not extinction, of the Corpus Christi procession was to be one of the features of Irish life in the latter decades of the twentieth century.

### 3. As the Council Approaches

Catholic life in Ireland at the end of the 1950s was rich numerically in terms of Mass attendance and the participation of people in devotional and popular piety. Following the conclusion of the Synod of Maynooth in 1956, when church legislation was reviewed, the Irish bishops commended the 'fidelity to religious duties', citing also the high attendances at the Holy Week liturgies following recent reforms which, inter alia, restored the traditional Easter Vigil (*Irish Press*, 16 August 1956).

However, as Fuller points out, because the various devotional practices and observances upon which Catholic life was being assessed involved obligation and repetition over quantified periods or times, many people missed out on the spirit of the devotion:

For many, the carrying out of these activities took on the nature of magical formulae and became somewhat compulsive. Many popular devotional cults such as the Nine First Fridays, the Five First Saturdays, the Thirty Days Prayer, and the Forty Hours devotion left themselves open to misunderstanding (Fuller 2002).

An example of this kind of spirituality was the tradition that pertained in Dublin whereby people would visit seven Altars of Repose on Holy Thursday. Another example cited by Fuller was the concern over when one was 'legally' late for Mass; in other words, at what point could one enter the church and still fulfil the Sunday or Holy Day obligation? Some experts cited the offertory, or so long as the person was present for the consecration and communion and also so long as that person another Mass for the parts that were missed (cited in Fuller 2002). Similar attitudes pertained to the practices surrounding fast and abstinence, all of which bore witness to the legalistic nature of Irish Catholicism in general as the Council approached.

Blanchard said of the forms of worship in the church in Ireland:

> The distinctive features in the forms of worship and religious devotion derive from the history of Ireland, and they include such trends as the unpopularity of Vespers, the relative unimportance of the liturgy, the brevity of sermons and the infrequency of High Mass. (Blanchard 1963)

There is no doubt that there was great piety among Irish Catholics, and this was deeply rooted in the memory of the people. However, the church life informing that memory, and the praxis from which it drew, was to undergo an *aggiornamento*, the reception of which would challenge the liturgical life of the Catholic Church in Ireland in the decades ahead.

## 4. The Council: Background and Irish Response

Vatican II was a council of reform for a world church. The council changed the way in which the Roman Catholic Church perceived itself and how it celebrated its major rituals. Officially, the emphasis had shifted from institution to the people of God, from the hierarchical and judicial to communion with Christ through the paschal mystery of his death and resurrection, a communion with others, rooted in the baptismal vocation of all believers. Flowing from these momentous changes were reforms of liturgy and governance that involved new ways of prayer, practice, and thinking.

A whole series of changes, particularly in the area of liturgy, had emerged in the local churches of Europe, notably in France, Germany, and Belgium, from the early decades of the century. These included a movement for liturgical reform, another which sought to promote Christian unity and a deeper ecumenical approach to relations between Christian churches, while a third called for biblical renewal and a greater use of history in seeking an understanding of scripture and the total ecclesiology of the church (Aubert 1954).

### 4.1. New Theological Currents

Much of the new theological thought at Vatican II revolved around *Ressourcement*, the return to sources of Christian tradition (scripture, the church fathers, and liturgy) (O'Collins 2012). Sometimes referred to as *nouvelle théologie*, *Ressourcement* challenged the dominant theological method of the pre-conciliar era, which was manualist and neo-scholastic. The ideas of ressourcement theologians dominated much of French theological thought during the period from 1930 until the 1960s. *Ressourcement* means going back to the sources of faith, such as the Scriptures, the teaching of the Church Fathers, and the understanding of the church being a community, rather than merely a repetition of something 'traditional' from the recent past. As Gabriel Flynn points out, 'the view of tradition proposed by the *nouvelle théologie*, far from being traditionalist, in the sense of a repetition of the recent past, was concerned rather with the unity of the ever-living tradition' (Flynn 2012). Such an approach was moving beyond the juridical model of faith that the Council of Trent (1545–1563) put forward, with a strong emphasis on orthodoxy and certainty. This was the model for teaching theology in seminaries and Catholic universities, including at Maynooth; it was rigid and, for the most part, without reference to the primary

sources. One Irish cleric who was familiar with *Ressourcement*, however, was Cahal B. Daly, a *peritus* at Vatican II and a future Irish bishop and cardinal. He followed with keen interest 'the many-sided revival which was in progress in the Church in France', adding that 'France was then leading the whole Church in terms of theology, liturgical renewal, pastoral innovation and evangelisation' and was the place where, in the words of Pope Paul VI, 'the intellectual bread of the Church is baked' (Daly 1998).

In the Ireland of the early 1950s, there were those who were prepared to tap into the new currents that were clearly emerging on the continent and to provide an outlet for discussion. One of these was the pastorally focussed journal, *The Furrow*, founded in 1949. The driving force behind the new venture was Dr J. G. McGarry, a member of the academic staff at St Patrick's College, Maynooth. *The Furrow* published a number of articles on aspects of the liturgical reform. Among these was one published in 1956 on baptism and the community by Mary Purcell, in which she compared the impoverished privatised celebration of the sacrament in Ireland with the communal celebration of the liturgy in Spain, as experienced by her. Coming as it did around the time of the changes to the liturgies of Holy Week, it was an attempt to highlight for the readership the potential and possibilities of the reform (Purcell 1956). A further important development in theology at this time was the launching of *Doctrine and Life*, a journal founded by the Dominicans in 1951, the first editorial of which placed it as 'the Irish counterpart of the French Dominican publication *La Vie Spirituelle*', which was one of those journals that provided a forum for theological insights in the years before the council (Ryan 1951). In time, *Doctrine and Life* was to be at the forefront in explaining and debating the impact of the conciliar decisions and vision. Yet another feature of the time was the rejuvenation of the *Irish Theological Quarterly* (ITQ), which had fallen into abeyance for several decades.

*4.2. Movement for Liturgical Reform*

The modern movement for liturgical reform dates to before the First World War and, over the years, featured such names as the Belgian Dom Lambert Beauduin, the German, Romano Guardini, and the French Dominican Pierre-Marie Gy. It was during the inter-war years, however, that the movement blossomed. The *Centre National de Pastorale Liturgique* was formed in France in 1943. That same year, a community Mass was sanctioned for use in Germany. This allowed for a mixture of Latin and German with greater use of the Dialogue Mass (Long 1950a). In 1947, these innovations received approval in the encyclical *Mediator Dei* of Pope Pius XII, which acknowledged the role of the laity in the liturgy. Shortly afterwards, the celebration of the Easter Vigil was restored to what liturgists see as its rightful place on the evening of Holy Saturday, initially on an experimental basis from 1951. In 1956, a whole series of reforms saw the universal re-introduction of the entire Easter Triduum, the three-day celebration by Christians of the paschal mystery, from Holy Thursday to Easter Sunday. This was a return to what happened in the early church, a return to what many see as the true meaning of tradition—*tradere*—a handing on. These changes in liturgical practice also included a call on the lay faithful 'to a more active and fruitful participation in the ceremonies', terminology that would resonate in the post-conciliar parlance with regard to liturgy. In essence, Vatican II drew from these developments to return the church back to its sources.[2]

During its early years, *The Furrow* devoted considerable space to liturgical renewal, enabling an assessment of the spiritual health of Irish Catholicism. Edward Long, a Donegal priest, bemoaned the excessive private piety of Irish people. As he put it: 'Private needs, the worries and hopes of the individual, dominate piety and there is little consciousness of being a family in Christ' (Long 1950b). In 1958, a Monaghan priest, Daniel Duffy, cited the strong Mass attendances and the improvements in the numbers receiving holy communion subsequent to relaxation in the fasting regulations, but highlighted that, particularly in rural areas, there was still 'the survival of a certain jansenistic attitude towards the reception of the blessed sacrament'. He attributed this to the 'false association of "confession and communion"' (Duffy 1958).

From the early 1950s, the Benedictine monks at Glenstal Abbey in Co Limerick became actively engaged in the promotion of liturgical reform. In 1954, they organised the first liturgical congress there and thus began the Irish liturgical movement. These congresses were to be a feature of Catholic life in Ireland for the next twenty-one years and they attracted some of the most high-profile scholars and thinkers in this area. The papers presented at Glenstal were often published in journals such as *The Furrow* and *Doctrine and Life*, together with commentaries on the progress of the congresses, thus enabling priests and others in parishes to gain some understanding of what was happening.[3]

The first Glenstal Liturgical Congress was devoted to Pope Pius XII's 1947 encyclical *Mediator Dei*. The encyclical set in motion a series of liturgical changes that were given their first expression in the 1951 restoration of the Easter Vigil and the 1956 introduction of the Easter Triduum (Acta Apostolicae Sedis 1955). The Congress was attended by seventy priests and contributors concentrated on liturgical developments and how they might affect urban and rural parishes. One of the speakers, the Irish Provincial of the Dominican Order, Fr Thomas Garde, told the congress that if Ireland was not 'liturgically minded, it is in turn great part due to the influence of the Penal Days' (*Limerick Leader*, 10 April 1954). The introduction of new Holy Week liturgies gave rise to what the Irish bishops described as 'an impressive demonstration of the faith and piety' with regard to attendances.[4]

One of Glenstal's promoters, Dom Placid Murray, saw the attitude of the majority of the clergy as the principal obstacle to reform, with many of them seeing liturgy as 'nothing more than rubrics or outward ceremonial' (Murray 1954). This was unsurprising in view of their training at Maynooth. As far as the laity was concerned, one of the senior bishops of the time had observed as far back as 1920 that:

> ... despite their pre-eminent piety, and devotion to Church practices, the strange fact remains, that there are no people who evince so much reluctance to active participation in Church functions as our Irish people. Their whole tendency in assisting at public devotions and liturgical functions is towards a devout passivity.[5] (McNamee 1920)

At an international liturgical congress at Assisi in 1956, the Austrian theologian Josef A. Jungmann reminded his listeners that the forms of the church's liturgy in the early centuries were determined by the necessity of bringing the faithful close to the act of worship. He pointed out that when the language used in the Mass, i.e., Latin, was no longer understood by the people, it gave rise to a 'petrification of the liturgy and its estrangement from the people'. He called for modern languages to be given a greater place in the rites of the church (*Irish Press*, 20 September 1956). The liturgical movement was further enhanced by this international gathering, which had Irish representation led by Bishop McNamee of Ardagh and Clonmacnoise, noted above. Other liturgical initiatives in Ireland included the introduction in February 1961 of a trilingual ritual (in Latin, Irish, and English) for the celebration of the sacraments and other liturgies (other than the Mass) in Ireland.

Nevertheless, despite the zeal and efforts of a small group of priests, the general attitude of the Irish bishops to liturgical reform in the pre-Vatican II period was lukewarm, as will be seen below. The work of the bishops' liturgy committee was so sparse as to be considered nominal. In fact, the liturgy committee and its role was regarded as so peripheral that in February 1961, Cardinal D'Alton, as chairman of the hierarchy, had to engage in a fact-finding mission in order to find out the names of its chairman and members. This incident arose from the arrival of a questionnaire from the Vatican II preparatory commission for liturgy.[6]

The Archbishop of Dublin, John Charles McQuaid, speaking to priests at their annual retreat in 1961, referred to his anxiety about any liturgical reform that might alter the spiritual life of people as follows:

To those who state that in Dublin there is no 'liturgical movement', as they call it, I should like to reply: generations of parish clergy, to whom is committed the pastoral care of

souls, and who have visitated and know their people, have slowly built up a framework of religious practice, within which the faithful have lived the Christian life with great fidelity.[7]

Was there stubbornness, denial, or even delusion in McQuaid's reaction? There is no doubt that he was aware of change happening. McQuaid quite clearly had reservations about the potential for change that *ressourcement* brought to liturgy and sought to delay it as long as possible. His attitude gave expression to a deep authoritarianism that marked Irish Catholicism, one that failed to recognise that biblical and liturgical renewal was very closely connected to a renewed interest in the church fathers. According to Roger Aubert, the aim of this renewal was to recover that which had been lost or neglected in the course of history (Aubert 1954). Such an intellectual approach to prayer and worship was not part of the fabric of Irish Catholic spirituality.

### 4.3. The Council Itself

The first stage of the council process was an ante-preparatory phase, beginning in the summer of 1959 when Rome sent a questionnaire to all bishops and to Catholic universities, inviting them to submit their concerns and ideas for discussion at the Council. This consultation, it was hoped, would provide an opportunity for those tasked with planning the council to understand the needs of the local churches and bring all into dialogue with the conditions of the wider world. All of the suggestions were synthesised into a series of questions covering eleven categories.

There were thirty-one individual ante-preparatory submissions from Ireland, twenty-four from diocesan bishops, two from auxiliary bishops, three from retired missionary bishops, and one each from the papal nuncio and St Patrick's College, Maynooth.[8] All the submissions were made individually, and there is no evidence of any consultation having taken place. Overall, they have been referred to as being 'short and skimpy', giving expression to 'a legalistic mentality and a concern to stick firmly to classical positions' and showing 'a scrupulous or timid conformism', especially concerning liturgy (Fouilloux 1995). Such a conformism, reflective of a peripheral mindset and a general lack of engagement with the strands of theological development on the continent, coupled with a hierarchical form of reception, was to inform much of the reception of the council reforms when it came to liturgy.

A survey of the Irish submissions reveals that they mostly dealt with clarity of doctrine and rubrics. Some bishops simply wanted the council to clarify or modify liturgical rules such as faculties or powers to delegate the administration of confirmation, faculties to erect stations of the cross, guidelines for penances for mortal sins, the length of intervals between the administration of *extreme unction*, the hours of the breviary, and related matters of rubrics. A number of bishops wanted clarity on regulations governing servile work, particularly on holy days of obligation, as well as reform of the rules governing fasting and abstinence.

Nonetheless, upon closer examination of the Latin texts, it is possible to detect some movement towards renewal and *ressourcement*. Cardinal D'Alton stated that any dogmatic definition should be christological; that Christ is the centre and fundamental truth of the Catholic faith and that the devotion of the faithful should be centred on him. He protested against placing Mary at the centre of devotion from the psychological and practical point of view (AD/I/II/III, p. 66). This submission was noted positively by the French theologian and peritus, Yves Congar (Congar 2012). Concerning liturgy, some bishops such as McNamee, Moynihan of Kerry, and O'Callaghan of Clogher called for consideration of the use of the vernacular in the context of people being better able to understand it. Bishop Staunton was positively in favour of its use in the Mass, adding that the epistles and gospels should be read in the vernacular only. Furthermore, he believed that the celebrant should face the people (AD/I/II/III, pp. 64–65, 68–69, 82, 86).

At the council itself, the contribution of the Irish bishops to the council sessions was, for the most part, sparse and defensive. During the one hundred and sixty-eight general congregations (as meetings of the council were termed) and amongst the many hundreds

of interventions, the Irish bishops made only eighteen. The first to speak was McQuaid, who on 24 October 1962 made a very short intervention of ninety-seven words during the discussion on the document on liturgy. He spoke on behalf of all Irish bishops, and his contribution simply asked that more emphasis be placed on the encyclical *Mediator Dei* and its references to the active participation of the laity in the Mass and to the Mass as a sacrifice (AS, I. pars I, i–ix, p. 414). Six days later, he again spoke on behalf of the bishops of Ireland, this time on Chapter Two of the same document. He emphasised the need for the faithful to acknowledge the sacrifice of the Mass as the centre of their religious lives. In relation to change, the Irish bishops wanted the Tridentine discipline to be rigorously maintained, and they were opposed to communion under both species. They also did not see the need for concelebration of the Mass, as was being suggested in the document. McQuaid said the people of Ireland were happier having several Masses celebrated on different altars at the same time (Carty 2007).

The Constitution on the Sacred Liturgy—*Sacrosanctum Concilium*—was adopted and promulgated on 4 December 1963. This was to be the foundation document for the renewal of liturgical practice and reflection in the years ahead and constituted the most immediate and visible sign of the council's reforms. The renewal of the liturgy was seen, therefore, not merely as a response to the liturgical movement alone, but as part of the total remit of the council, as part of both *ressourcement* (in liturgical terms) and *aggiornamento* (in terms of updating to speak to the modern world). The central importance of liturgy in the life and mission of the church is summed up in paragraph 10 of the document which states: 'The liturgy is . . . the high point towards which the activity of the church is directed, and, simultaneously, the source from which all its power flows out'. This put worship and liturgy, and especially the eucharist, at the forefront of the church's life.

While all the Irish bishops present in Rome voted for the adoption of the final draft of the constitution, they had earlier submitted several *modi* or amendments to the initial draft. A few of these were notable insofar as they reflected the cautious and protectionist approach to change, a clerical mindset, and a lack of significant theological insight into what was actually happening at the council. Among these was one which sought the addition of a clause that would prevent the canon of the Mass (or Eucharistic prayer) being recited in the local language.[9] In relation to the re-introduction of the historic practice of concelebration and the mandatory provision for its use at the Chrism Mass on Holy Thursday and on other occasions, several Irish bishops inserted a *moda* seeking to have that permission restricted to the bishop of the place and not to the bishops of the country as a whole.[10] This amendment, which did not make its way into the final document, reflected the centuries-old territorial mindset of Irish bishops and their vision of themselves as 'rulers' rather than as co-workers. The exclusively clerical outlook also permeated another unsuccessful amendment that was proposed, this time to article 79 of the liturgy document, concerning sacramentals. That article states that provision could be made for suitably qualified lay people, with the permission of the bishop of the area, to administer some sacramentals. They included blessings, consecrations, and dedications. However, a number of the Irish bishops petitioned unsuccessfully to have this provision deleted.[11]

## 5. Implementing the Council

The work of turning the council's vision into a reality for the local churches in Ireland began with the establishment of a liturgy commission of the Irish hierarchy and the institution of formal decision-making processes on the degree to which the vernacular would be used. The translation work was undertaken by a new international body, the International Commission on English in the Liturgy (ICEL). This was set up in the weeks prior to the promulgation of *Sacrosanctum Concilium* in 1963 by the bishops of fifteen countries where English was spoken. Ireland's first representative on this body was Joseph Walsh, the Archbishop of Tuam and chairman of the liturgy commission. He was replaced in this role in 1969 by Bishop Michael Harty of Killaloe. ICEL translated the Latin editions of the liturgy books and liaised with local conferences of bishops, some of whom were

collaborating to work out common interim translations. Once translations were approved by the hierarchy, they were sent to Rome for formal confirmation before being introduced into the liturgy.

In June and October 1964, the Irish bishops decided formally on which Mass prayers and responses would be in the vernacular, and these were submitted to Rome for approval. On 8 November, they announced in Rome their decision to introduce the vernacular into certain parts of the Mass. It was noted that the changes would be introduced in stages. In addition to the Epistle and Gospel, the prayers at the foot of the altar, the *Kyrie, Gloria, Credo, Orate Fratres, Sanctus, Pater Noster, Agnus Dei*, and *Ecce Agnus Dei* would be recited in the vernacular. These initial changes came into effect in March 1965.

On Sunday 7 March 1965, for the first time, Catholics throughout Ireland celebrated Mass with the responses in the vernacular. Teilifís Éireann, the national broadcaster, hosted a live broadcast of Mass in Irish, celebrated by Cardinal Conway at the Franciscan College in Gormanstown in County Meath, which was attended by President Éamon De Valera. In Dublin, the new liturgy received an enthusiastic response, with one priest saying that people had been 'thrilled' to take a vocal part in the Mass. Masses were celebrated at the Pro-Cathedral from 6.30 a.m., and additional priests were required for the distribution of Holy Communion. At St Andrew's Church, Westland Row, additional priests led the people from the pulpit in answering their responses (*Irish Press*, 8 March 1965).[12] The *Irish Independent* reported of people saying they had 'a greater sense of union with the celebrant', with one pointing out that the word 'assist' now had a fuller meaning. Others mentioned that 'their minds now showed less tendency to wander'. The same paper reported people in Kilkenny responded so positively that priests were 'left wondering why this change should have delayed so long' (*Irish Independent*, 8 March 1965).

While the above shows an acceptance of the changes, comparative analysis with other countries shows that the Irish bishops were much slower and less enthusiastic in implementing the changes than some other English-speaking countries. An indication of the warmer reception of liturgical changes in other countries was the fact that the South African and Australian hierarchies had their petitions for even greater use of the vernacular submitted to the Holy See in March 1964, three months before the Irish hierarchy had even had its first vote on the question.[13] In fact, the Irish bishops had sight of the South African and Australian petitions before their June meeting when they voted on their limited initial petition.[14] The bishops of England and Wales had accepted the liturgical changes and made their petition in plenary session on 24 April 1964.

By the first Sunday of Advent 1966, new interim texts for many of the prayers and responses were introduced following agreement by the bishops' conferences of Ireland, Scotland, England, and Wales. In May 1967, the vernacular was extended to the canon of the Mass (the eucharistic prayer), something that the Irish bishops had resisted at the council. The work of revising the texts and format of the Mass continued in Rome until 1969 when a new Roman missal was approved by Pope Paul VI. This was promulgated for the entire church in March 1970 and following translation by ICEL, the English language version of the Roman Missal came into effect in Ireland on 16 March 1975. The Irish bishops unanimously approved the ICEL translation of the *Ordo Missae* by secret ballot at a meeting on 7 and 8 October 1969. This *Ordo* was incorporated into the new *Roman Missal*, which was approved for use from 1975. At the same meeting, a recommendation was agreed for uniform practice in Ireland with regard to the posture of the laity at Mass.[15] However, most notable among other changes were the posture and position of the priest during the Mass. Previously, the priest had celebrated the Mass with his back to the congregation, whereby the people offered the sacrifice of the Mass through him. Following the council, the renewed emphasis on the eucharistic assembly of priest and people led to the priest facing the congregation, as a presider and celebrant. This would be given greater effect through changes to the architectural layout of churches.

An important area of renewal was in the domain of music during the years following Vatican II. More efforts were also made to encourage greater participation by congregations

in singing at Mass and other liturgical events, especially at parish missions or at places of pilgrimage such as Knock in County Mayo and St Patrick's Purgatory at Lough Derg in County Donegal. Reflecting on the early years of his ministry on Lough Derg, Laurence Flynn notes that the improvements in music there in the years after 1978 must have been 'a reflection of something that was happening rather imperceptibly throughout the country, both at the level of repertoire and in terms of a growing sense of what is good liturgical practice' (Flynn 1991). Indeed, liturgical practice at Lough Derg did reflect the changes that were happening. A survey of the Lough Derg archives shows that the reception of the liturgical changes there mirrored much of what was happening elsewhere. The Dialogue Mass, at noon on Day 2 of the pilgrimage, was introduced in 1961. In 1970, an evening Mass was introduced, at the insistence of Bishop Patrick Mulligan of Clogher. From 1970 too, the celebration of Stations of the Cross included greater emphasis on scripture. The 'Instruction' following early morning Mass was discarded in 1977. In 1980, the then Bishop of Clogher, Joseph Duffy, had the renewal of baptismal promises introduced as part of the exercises, reflecting the centrality of pilgrimage and conversion alongside the foundational emphasis placed on baptism and reconciliation by Vatican II, thus linking 'the two sacraments together as sources of renewal in the Christian life' (McGuinness 2000).

In May 1967, the Holy See issued a further instruction regarding the liturgical reforms entitled *Eucharisticum Mysterium*. It concentrated on the eucharistic mystery of the Mass and various theological and liturgical principles with which priests and lay people should be acquainted. Among these principles were the connection between the Liturgy of the Word and the Liturgy of the Eucharist, the nature of active participation in the Mass, and the different modes of Christ's presence in the eucharist. It recalled the teaching of *Sacrosanctum Concilium* that Christ is present in the liturgical celebrations. It reiterated the conciliar teaching that Christ is also present in the faithful who gather for a liturgy and in the proclamation of the scripture. Further, Christ is also present in the person of the priest and, above all, in the species of the eucharist (SC 7). This was new territory for Irish Catholics used to devotional practices that emphasised the presence of Christ exclusively in the blessed sacrament.

*5.1. Material Effects*

The various changes necessitated a plethora of texts being provided for use by both clergy and laity alike. Cards and leaflets in Irish and English were provided at church doors for people (*Irish Independent*, 1 March 1965), and this increased as the changes unfolded. Louise Fuller cites examples of priests using temporary leaflets until the completed missals became available—a period and practice that in clerical parlance became known as a 'paperchase' (Fuller 2002). Other developments that involved material items included the introduction of parish newsletters or bulletins in place of the weekly 'announcements' by the priest. By the late 1970s, these were common in most parishes, and texts often included the Scripture readings of the Mass, thus aiding greater reception of the Word of God. In 1965, the Dublin publishers Eason and Sons published the first prayer book for young people, incorporating the new English text of the Mass. It had two sections, one a method of following the Mass for the very young—with colour illustrations—and the other a section for teenagers. There was also a section dealing with the liturgical year (*Westmeath Examiner*, 27 November 1965). In 1970, a Catholic Diary, including the new texts of the Mass, was published by Mount Salus Press. It also included prayers before and after Mass, composed by Fr T F Brophy, Carlow (*Evening Herald*, 23 December 1969). In 1977, with all of the new rites now approved and in place, the Diocese of Clogher produced a *Mass and Prayer Book* for use in the parishes of the diocese. Published by Shanway Publications, it contained the Order of the Mass, the Proper for each season and holyday, and the rites of the sacraments together with a selection of devotional and other prayers, including prayers for each day, birthdays, wedding anniversaries, and other occasions. It also contained prayers in Irish and Latin as well as hymns. In the words of Bishop Patrick Mulligan, it was provided to 'help anyone who uses it to enter more fully unto the liturgy and the rites of the Church'[16].

*5.2. Gradualism: A Slower Approach*

The slower approach of the Irish bishops may have reflected a generally cautious attitude toward change in the wider church in Ireland. People like Conway realised that further changes were inevitable and, therefore, a gradual approach was required. Bishop Browne, at a diocesan conference in February 1964, asked priests to report to him concerning whether or not certain parts of the Mass should be addressed to the people in their local language and whether people really wanted any changes at all. A survey of some of the findings shows that while some people favoured changes such as the epistle and gospel being read in the vernacular, there was a reticence on the part of many others towards change. Of course, whether the priests were reporting in such a manner as to coincide with own views or perhaps expressing the outcomes in a way they thought the bishop would want to hear is another matter.

Some examples of a cautious approach to change are referenced in the archives of the Diocese of Galway. The parish priest of Castlegar, Joseph Mitchell, reported that he and his curate had interviewed one hundred and ten 'heads of families' in their homes and that sixty-four of these wished for no change, while forty-one wished that English would be introduced and five wanted the introduction of Irish.[17] The parish priest of Kilfenora, County Clare, James Horan, reflected a narrative that was common among a significant body of mostly older clergy: that liturgical participation was best achieved through simplicity. He said that the majority of his parishioners wanted no change, adding, 'Few country people will venture an outright statement but more than one might imagine [text unclear] associate themselves very closely with the priest by the use of missals and simple prayer books'.[18] In the Galway city parish of *An Cladach*, Damian Byrne reported that people were, in general, 'suspicious of change for the sake of change' but that any change which would help them attend at Mass with a better disposition would be welcomed. He then went on to recount the response of someone who had participated in liturgies outside of Ireland that used the vernacular.

> The most informative discussion I had was with a docker. He would like all the parts of the Mass at present sung by the choir to be said "in English" by the priest and people. He spent some time in England and was very impressed by, for example, the recitation of the Apostle's Creed by the people during the recitation of the Credo by the priest.[19]

On the other hand, writing from Gort, Co. Galway, in March 1964, Denis Hynes, chairman of the Galway Diocesan Liturgical Commission, stated that the liturgical movement in England amounted 'almost to an agitation' and did not appeal to Irish Catholics. As he writes:

> Irish Catholics knew and appreciated what the Mass essentially was . . . and though they may not have had a clear understanding of the meaning of the prayers or of the symbolism in the Mass, their act of faith embraced the whole liturgical celebration.[20]

Overall, most laypeople were reported as seeming to be open to, but not excited by, possible or even imminent liturgical change. Thus, we see here, on the one hand, the lack of any liturgical tradition in Ireland, compared with what was happening elsewhere, and, on the other, the mixed degrees of openness to reform, other than on the part of those who had witnessed liturgical renewal elsewhere. Others were satisfied to leave to the hierarchy to be the receivers for them.

The early stages in the process of agreeing translations for the Mass and the scripture readings highlights the mix of caution, inertia, and even narrow nationalism of some of the church leadership in Ireland and their inability to grasp fully what was happening in the wider church in terms of liturgical change. It also highlighted that Ireland was no longer in control of its own liturgical destiny. Outside influences, particularly from America, would come to the fore. The influence of the American church in the translations during the period that led to the new English-language missal in 1975 is evident. ICEL had

its headquarters in Washington DC and was heavily staffed and influenced by American clergy and liturgists. Other questions to be dealt with included the ambiguity of language in the conciliar documents, the need for compromise, and the desire to bow to modernity in the quest to achieve accessible language. These would be the dominant factors that would inform the reform of liturgy in Ireland during the decade following Vatican II. Nonetheless, where the Irish bishops had the power, they used it cautiously to ensure that changes were introduced gradually. It is clear that, for the most part, the Irish bishops, in common with others in the English-speaking world, were content to leave the bulk of the translation work to the Americans and thus allow them shape the wordings of prayers and responses that would become part of liturgical life in the decades ahead. While the translation of the Mass into English was ongoing, the production of a missal in Irish came to fruition in 1973, ahead of its English counterpart. The introduction of the vernacular provided an opportunity for supporters of the Irish language movement to lobby for and promote the use of Irish and the promotion of Irish-language Masses (see Bruce 2016).

The change in the language used at Mass, from Latin to the vernacular, was perhaps the most outward sign of change after the council. Advocates of the council and those who preferred a more traditional approach made this area of church life a form of battleground. A Co Waterford 'traditionalist' writing in the Irish Catholic in March 1966 called for Latin Mass to be an option in parishes, adding, rather erroneously, that 'St Patrick made Latin the language of the Church in this country. Since then the Latin Mass has been adequate to inspire countless vocations and countless nameless martyrs to defend it' (*Irish Catholic*, 10 March 1966). This drew a response from a woman in Co Longford, who replied 'I think the English is wonderful', adding that she loved the re-introduction of the Prayer of the Faithful and the inception of evening Masses (*Irish Catholic*, 24 March 1966). A Dublin reader described 'a period of shock' following the introduction of the changes, but concluded that 'the Fathers in the General Council, under the guidance of the Holy Spirit, were those with the necessary information and authority to decide what was best', adding that this principle helped in accepting and benefitting from the liturgical changes (*Irish Catholic*, 24 March 1966). In May 1969, a commentary in the *Cork Weekly Examiner* noted that the experience in the Dublin archdiocese generally was that Masses in the vernacular had 'aroused a wonderful upsurge in devotion among congregations, large sections of which (Sunday being the morning-after-the-night-before) would previously have been passive non-missal carrying attenders' (*Cork Weekly Examiner*, 8 May 1969). This provoked a response from an English reader who saw the observation as 'a terrible indictment of the Irish' and added she would have thought 'the largely silent Latin Masses of the past would have suited those with a hangover better than reciting Mass in English' (*Cork Weekly Examiner*, 22 May 1969).

A letter writer to the *Evening Herald* in December 1966 questioned 'the delay in some places in bringing the vernacular into use for the Service called The Absolutions at the coffin after Requiem Masses', noting that the prayers were still being said in Latin in some areas of the country (*Evening Herald*, 29 December 1966). The new Funeral Rite, with prayers in the vernacular, was introduced on an interim basis in 1969 and further revised in 1974 and again in 1993.

*5.3. Lay Participation*

The new place of lay participation was given further emphasis in the creation of new ministries, notably that of reader or lector and, later, that of extraordinary ministers of holy communion. Other roles, such as those of cantors and musicians, would also come to be seen as ministries that shared in the liturgical life of the parish. An early indication of the reception of this change was the newsworthiness of laypeople reading the Scriptures at Mass, such as when the Taoiseach Seán Lemass read the epistle at the consecration of the new Bishop of Waterford and Lismore in December 1965. He became the first lay person to participate in such a liturgy, while the Mayors of Waterford and Clonmel took part in the offertory procession (*Irish Press*, 19 December 1965). However, not every first occasion

for the introduction of lay ministry was as high profile, nor were the ministers. In fact, it would be some years before formal support and recognition for the role of lay reader was in place. Further, the reception of the liturgical changes was not only hierarchical, but it was gendered too. At the annual retreat of his priests in 1968, McQuaid gave 'a very grave warning' that he would not tolerate any interference with the regulations of the Holy See, and included in his assessment 'no reading of lessons by girls or women (as has happened); no Offertory processions of nuns or others; no private observations'.[21] Equally punctilious about observing regulations from the competent authorities was Bishop Browne of Galway, who, in 1970, in an exchange of correspondence with Cardinal Conway, questioned whether it had been agreed by the Episcopal Conference that women were permitted to read at Mass. Conway replied by quoting the relevant extract from the November 1969 meeting of the conference, that 'it was agreed that women be allowed to read Scriptural readings other than the Gospel where the local Ordinary judges this to be opportune'.[22] Here, once again, we see an adherence to power through regulation. When compared to more advanced countries such as Canada, where the French-speaking influence was more manifest in driving liturgical change, Ireland was once again lacking. For example, from March 1964, the reading of scripture in the vernacular at all Masses in Canada was the norm. This was a year in advance of the Mass responses being made in the vernacular (Dias 2011).

Two other developments honoured the active participation insisted on by *Sacrosanctum Concilium*—the introduction of Eucharistic ministers (or Extraordinary Ministers of Holy Communion) and the option to receive communion in the hand (*communio in manu*). In relation to extraordinary ministers of Holy Communion, these were introduced in France by indult dated 13 March 1970 (Brulin 2011). The US bishops requested the Holy See for permission to introduce them in the USA in 1971, and this was granted. By virtue of the Roman document, *Immensae Caritatis* in January 1973, this was extended to the universal church from that date. In Ireland, while there is evidence of it having been discussed at a meeting of the Council of Priests in Galway in 1974, where a proposal to have it considered was passed unanimously, it would be the late 1970s to mid-1980s before Eucharistic Ministers became a reality in most parishes.[23] Recognising that this was a very significant step for some people, parishes made provision for choice, as can be seen by a notice in a parish bulletin in a Dublin parish in 1981 that stated that 'some people find it strange to receive the Blessed Eucharist from the hands of a lay person', adding that 'a person has every right to receive the sacraments from whom they wish' (Fuller 2002). Consideration of the reception of communion in the hand was first raised by another Roman document *Memoriale Domini* in May 1969. It sought the views of the bishops of the world. The French bishops were once again very positive and requested an indult (Brulin 2011). However, a perusal of the minutes of the Irish hierarchy shows that the matter was discussed first in June 1973 and, again, in 1976. It transpires that at the 1973 meeting, a proposal to apply to the Holy See for an indult allowing the introduction of communion in the hand was rejected by fourteen votes to twelve. Immediately, however, a second proposal to apply to the Holy See for a similar indult, in the event of the hierarchy of England and Wales having agreed to do so, was carried by twenty-one votes to four. On 29 March 1976, Cardinal Conway wrote to all Irish bishops informing them that the English and Welsh Conference of Bishops had received the required indult and that he had asked them to delay any announcement so that the Irish bishops could have an opportunity to formally decide to proceed with the indult process. This was agreed at the April meeting of the Standing Committee.[24]

An area of notable change was that of church art and architecture. Like the liturgy itself, church architecture was influenced by changes that originated in continental Europe following World War One. While few of these impacted Ireland, there was sufficient interest in this area by 1955 when a well-attended conference on church architecture was held in Dublin, under the auspices of the Royal Institute of the Architects of Ireland. The new liturgical reforms called for adaptation in terms of space usage so as to maximise the levels of participation by those gathered—priests and laity—and to emphasise the role of the

assembly in the liturgical actions. In response, the Irish bishops established an advisory committee on sacred art and architecture in 1964. It was chaired by J.G. McGarry, editor of *The Furrow*, and included a number of architects and artists. In June 1966, directives were issued on the building and re-ordering of churches, arguing that 'participation of the faithful can best be achieved in a church which has been properly planned or re-organised' (Irish Episcopal Commission for Liturgy 1966)[25]. Among the changes was the new place of scripture and the preaching of a homily in the Mass. This allowed for the introduction of an ambo and the abandonment or replacement of pulpits. Another feature was the removal in many churches of the altar rails that divided the sanctuary area from the remainder of the church. It was almost thirty years later before a comprehensive and more definitive pastoral directory covering this area was published in 1994 (Hurley 2001). The extraordinary elapse of time before the publication of a definite pastoral directory in this area is yet another indication of the piecemeal approach to the reception process in Ireland with regard to Vatican II. However, several new churches emerged in the period after the council and reflected the developing understandings of the conciliar vision while perhaps also reflecting a lack of due regard for balance in terms of heritage.

Perhaps the biggest undertaking in terms of church building in Ireland at this time was the completion of the new Cathedral of Our Lady Assumed in Heaven in Galway, which was completed and dedicated in August 1965. The plan of the cathedral that emerged was 'conscious of Galway's long ecclesiastical past' and thus a varied one with a mixture of designs from the classical and renaissance periods, such as the dome, pillars, round arches, and the various-shaped windows (O'Dowd 2015). Delivering his annual report on the progress of the cathedral in 1964, Bishop Browne noted that it was 'of interest to know that the cathedral fulfils perfectly the requirements of the new Liturgy Constitution for the architecture and design of churches and altars'.[26] The cathedral was nearing completion when *Sacrosanctum Concilium* was adopted, and a survey of the building project papers, held at the Galway Diocesan Archives, shows that there was no demand for any re-ordering of the cathedral in the light of the liturgical reforms. However, and as architect Richard Hurley contends, when the architect John J. Robinson was commissioned to design the cathedral in 1949, 'Vatican II was not even on the horizon', and it was largely completed by the time *Sacrosanctum Concilium* was adopted; it cannot therefore be seen as in any way responding to the liturgical changes (Hurley 2001). One of the first renewal projects in the north, St Brigid's church on Derryvolgie Avenue in Belfast, was re-opened after extensive renovations in 1965, at a cost of £30,000. It was the first church in the Diocese of Down and Connor where the altar was positioned for the priest to face the congregation. The original altar, which was awarded first prize for carving at the 1908 Paris Exhibition, had the *reredos* removed. St Patrick's church in Dungiven was the first church in Diocese of Derry to conform to the liturgical reforms, allowing for the celebration of Mass versus populum. The first newly-built church in the whole island to have no pulpit or communion rails was in Ballyjamesduff, County Cavan, which was dedicated in October 1966 (*Anglo Celt*, 14 October 1966). Across the country, a number new churches were built during the decades following Vatican II, and the new designs reflected the role of the assembled community of priest and people gathered together to celebrate eucharist.

However, it is fair to say that the initial reception of modern architectural and artistic designs gave way over time to a more restrained appreciation. This was added to by civil legislation, such as the Planning and Development Act 2000, which included provisions designed to protect churches as heritage spaces and structures. While the post-conciliar period in Ireland saw some notable architectural and artistic creations, the place of Catholic heritage and tradition is an aspect that was neglected in the endeavour by some to, in the words of the church artist Ray Carroll, get 'involved in the spiritual movement of the Church in modern times' (*Sunday Independent*, 23 November 1975). An example of the inclusion of heritage and tradition in the conversation is the outcome of the second re-ordering in St Patrick's Cathedral in Armagh in the early 2000s. This included the provision of a new altar, an ambo and presider's chair, together with the reinstatement of

the Marian chapel, the creation of a new and more open rood screen, the retention of the sanctuary lamp, and the refurbishment of the *reredos* of side-altars. This has been described as an 'authentic renovation' that 'honours the past and the present together so that any new work is a valid contemporary expression of the worshipping community' (O'Hare 2006). Here is an example of a meeting of memory and theology, a coming together that follows not a hierarchical reception but, rather, reflection, experience, and prayer. In this way, the noble simplicity espoused by the council can sit side by side with tradition, connecting with the past 'but not to uncritically reproduce it' (O'Hare 2006).

## 6. Popular Reception of the Council

Reception is a relatively new phenomenon in theological discussions. Reception concerns the process by which a faith community receives changes to beliefs, practices, norms, and laws. It may be defined as 'the process through which an ecclesiastical community incorporates into its own life a particular decision, teaching or practice' (Rausch 1987). It involves the whole Church. In *Lumen Gentium*, Vatican II emphasised that all the baptised, as members of a Christian community, share in Christ's prophetic office and, as the people of God, they 'have an anointing that comes from the holy one . . . shown in the supernatural appreciation of the faith (*sensus fidei*) of the whole people, when . . . they manifest a universal consent in matters of faith and morals' (LG, 12). The '*sensus fidei*', understood as making sense of the faith and giving assent to it, goes to the origin of the church and how believers, as a faith community, share and practice their beliefs. Ormond Rush refers to the 'corporate ecclesial sense as the *sensus fidei fidelium*, the sense of the faith of the faithful'. This gives rise to the question, what is a receiving community? According to Őrsy, a receiving community 'is the principal agent in the process of learning and receiving'. Such a community must be of one mind and one heart. Further, the act of receiving cannot be undertaken exclusively by the ecclesiastical authorities or their representatives. It must involve the whole people, with at least a significant majority being in favour of it (Őrsy 2009).

These are important considerations when we look at the reception of the liturgical changes in the church in Ireland. Who were the receivers and what informed the reception?

While, in general, the liturgical reforms were implemented over a relatively short time span, they did not create any significant upset in the Roman Catholic Church in Ireland. The caution with which the Irish bishops approached the implementation at the early stages of the translation process may have had more to do with their lack of historical and theological training in the area of liturgy, and their lack of awareness of *ressourcement* in particular. The translation work was left to ICEL, and the reforms were, in the main, implemented dutifully in accordance with Roman norms.

Did Catholics in Ireland find life and relevance in the celebration of the new liturgy? Did a liturgy that was celebrated in the vernacular and that encouraged and facilitated participation speak to modern people and did people see the liturgy as 'the high point towards which the activity of the church is directed' and 'the source from which all its power flows out' (SC, 10)? To try to answer these questions, one can only point to commentaries by various observers in the field of liturgy. P.J. Brophy of St Patrick's College, Carlow, stated that 'the new liturgy seems to have little to say about tomorrow since it is presented in the categories of yesterday' (Brophy 1974). Taking into account the bigger picture of Irish society at the time, Fuller (2002) states:

What the council could not absolutely foresee . . . was the huge impact of the consumer society, pop culture and communications revolution and a more individualistic age which would lead people to have far higher expectations of what was meaningful for them. The irony was that, within a decade of the close of the council, Irish society and culture were so utterly transformed that many observers in the 1970s felt that, radical though the changes had been, and notwithstanding the traditional devoutness of Irish Catholics, the renewed liturgy did not really speak to their needs.

A similar point was made by Eamonn Bredin, a lecturer in sacramental theology at Mount Oliver Institute for Religious Education in Dundalk. Writing in *The Furrow* in 1979,

he stated that the liturgical reforms seemed to 'have left the inner core of people's lives untouched'. In reaching this conclusion, Bredin pointed to a series of studies of religious beliefs and practice that had taken place in Ireland in the 1970s. For example, the survey carried out in 1973–1974 by the Council for Research and Development found that while there was a 91% rate of weekly Mass attendance, other findings showed cause for concern, especially a rate of just 76.5% Mass attendance among single males (Fuller 2002). Bredin referenced a statement by the National Conference of Priests (in May 1979) that 'the main danger to religion in Ireland is not unbelief but shallow belief, a convention retained but only on the margins of life, a religion without challenge and without depth' (Bredin 1979).

An example of the changing landscape of religious practice was reflected in comments by Conway in 1970 when he criticised the poor attendances at the Holy Week liturgies in Ireland. Speaking in Moy, Co Tyrone he noted that for some people 'Holy Week is just the week before the Easter holidays' and that many people were content just to attend Mass on Easter Sunday morning.[27] When this comment is juxtaposed with the statement by the Irish bishops in 1956, already referred to, concerning the large attendances at the Holy Week liturgies as evidence of the practice of the faith and the reception of those reforms, this expression signalled that a significant shift was underway.

The quality of the celebration was also a consideration. Among the factors informing this element of reception were the introduction of the vernacular and the homily. Both of these made communications skills an important component of the relationship within the liturgical assembly. Priests were now expected to be transformed from silent actors to active and vocal leaders and animators of a gathered community. It meant that Mass had to be meaningful for people. A submission from a priest to the special meeting of the Irish bishops at Mulranny in April 1974, a meeting specifically to review progress since Vatican II, put it starkly:

The vernacular has put us in the dock. Unless the people find Mass meaningful they may not continue to go to Mass. To say that they are lacking in faith, that they are not putting anything into the Mass etc is not the answer. The pedagogical value of good celebration is written all over the documents of the magisterium since Vatican II. Good celebration will help to counteract the trend away from Sunday Mass (Irish Episcopal Conference 1974).

In response, the bishops highlighted elements of non-reception of the liturgical reforms, which they termed as 'abuses in the name of tradition' These included non-cooperation with liturgical renewal, failure to give a homily, and failure to provide a worthy setting for the liturgy. They also called for greater attention to the length of Masses.

It must be acknowledged that after 1965, people had become increasingly accustomed to, and came to insist on, the participative elements of the liturgy. This was most visibly seen at family and community-related celebrations such as marriages and funerals. Priest and liturgist, Edward Magee of Down and Connor, states that 'while exterior manifestations of participation facilitate greater involvement in the liturgy, an interior disposition is equally important'. In implementing the council's liturgical changes for a laity that was primarily used to seeing the Mass as an obligation to be fulfilled through quiet observance, there was little or no attention paid to the theological and catechetical dimension of those changes. The hierarchical reception of the council in Ireland meant that the import of the changes, however understood, was lost on the majority of people. As Magee notes, 'inculturation of the liturgy and creativity in expression was achieved at the expense of a more interior and spiritual sense of participation in the liturgy' (Magee 2015). Such an assertion relates to the personal rather than communal reception of the reforms, an aspect which is always difficult to quantify.

However, liturgical formation of the laity and the clergy alike was slow. With few exceptions, little or no discussion on the changes was held in public within dioceses or parishes in terms of how the liturgical reforms represented the action of the council. As Bredin puts it, 'all they [the people] were ever told was that there were new changes and that these were the new responses' (Bredin 1979). This is not to say that no positive actions

were taken. The creation of the Institute for Pastoral Liturgy in 1974 and the appointment of Seán Swayne as its first director enabled considerable outreach in terms of pastoral liturgical training, including in the area of church music.[28] Between its formation and 1982, it had provided a one-year course to over two hundred people, and through its shorter course, seminars and study-days had touched many thousands more (Swayne 2015). Speaking at a liturgy conference in Rome in 1982, Swayne reviewed progress in this area since the council and stated firmly that 'to hope for liturgical renewal without prior liturgical formation is futile', adding that the difficulties in putting the liturgical reforms into practice after Vatican II stemmed from the fact that 'neither priests nor people have received an adequate liturgical formation'. Swayne held the view that liturgy itself, well-celebrated, becomes a school of deep formation. However, enabling that to be realised in parishes and communities would remain a challenge for decades to come.

The liturgical reforms and developments of Vatican II sought to deepen the significance of the encounter with the God and to enrich people's experience of it. It did not seek to eliminate the devotional life of the Church. Speaking at the end of a Corpus Christi procession in Cork in May 1970, Cardinal William Conway was mindful of the memory of the people and the yearning to establish contact with the transcendental when he described as a great mistake any attempt 'to eliminate from the devotional life of the Church everything that might not suit the taste of a fastidious middle-class person—to create a liturgy for the bourgeoise as it were', adding that similar mistakes were made in the sixteenth century, which the church has always avoided.[29] In making such a declaration, Conway was affirming that the liturgical reforms did not change what the Mass was- namely, a perpetuation, a making present again, of the sacrifice of Calvary. The priest, he said, was not just a presiding officer at a community assembly but a person invested with sacred and supernatural powers.

As the 1970s progressed, however, the frequency of traditional devotional rituals and practices reduced, as did the visibility of groupings such as confraternities and socalities. Writing in 1982, the then Archbishop of Dublin, Dermot Ryan, noted that popular devotions 'have declined considerably since the introduction of Evening Mass. There has been a tendency to celebrate Mass on every possible occasion when perhaps some other devotional exercise might be more appropriate' (Ryan 1982). Ryan added that there was evidence of some balance being restored, citing the increase of Eucharistic adoration in some parishes and the emergence of prayer groups arranging services around the Word of God rather than the Eucharist, with some parishes even on occasions celebrating the Liturgy of the Hours.

Prayer in the home, a cornerstone of piety and faith in Ireland, saw a decline in the years following the council, due mainly to the increased materialism and secularisation of Irish life. Conscious of the rich tradition of prayer in Irish life and culture, Pope John Paul II on his visit to Ireland identified this as a challenge. Speaking in Limerick on the last day of his visit to Ireland in 1979, he declared:

> Ireland in the past displayed a remarkable interpenetration of her whole culture, speech and way of life by the things of God and the life of grace. Life was in a sense organised around religious events. The task of this generation of Irish men and women is to transform the more complex world of modern industrial and urban life by the same Gospel spirit. Today, you must keep the city and factory for God, as you have always kept the farm and the village community for him in the past.

He concluded by asking families to renew the commitment to family prayer.

> Your homes should always remain homes of prayer. As I leave today this island . . . , may I express a wish: that every home in Ireland may remain or may begin again to be, a home of daily family prayer. That you would promise me to do this would be the greatest gift you could give me as I leave your hospitable shores. (Pope John Paul II 1979)

The papal visit was followed by the publication on St Patrick's Day 1980 of a Pastoral Letter from the bishops of Ireland on the topic of family prayer and the future of faith generally. In addition to dealing with prayer in the home, new catechetical developments, questions concerning parents and teenagers and justice issues—as well as reminding people that 'prayerless homes' would in the long run 'bring about a Godless people'—it dealt with the question of religious education for all, asking '[h]ow "grown up" really are we in what concerns religion?' It challenged Irish people to reflect on their understanding of religion and whether it was more suitable to primary education level than to modern adult life. They went on:

The primary source of education in the faith is the Church's liturgy. Trying to improve our understanding of the liturgy of Mass and the sacraments, and to participate more intelligently and actively in them, is a most effective form of religious education. The readings and the homily have a leading part to play in the process of religious education (Irish Episcopal Conference 1980).

The bishops noted that they and priests were 'powerless and helpless' without the support, loyalty and help of the laity and of parents in particular, adding that the task was a common one—not to weaken or endanger the link in faith between the generations of Irish people. While there were undoubtedly numerous efforts on the part of many dioceses and parishes, along with schools, to encourage prayer, the challenges posed by secularism and materialism continued to grow in the 1980s and 1990s.

*Ecclesial Movements*

Among the significant developments at the international level after Vatican II was the emergence of new movements, or new ecclesial movements as they were termed. Most of these began in Europe and had founders who were young. Each of them brought a characteristic spirituality that helped people to renew and live out their faith in the world today. Until Vatican II, the dominant Catholic movement had been the Legion of Mary, founded in 1921 by Dubliner Frank Duff. He had a great foresight in terms of his vision for the role of laity in the church, combining prayer and social action. The movement spread to all corners of the earth and Duff himself was invited to be a lay observer at Vatican II, where his arrival evoked a standing ovation from all in St Peter's Basilica. However, the Legion of Mary did experience some opposition in Ireland from, notably from Archbishop McQuaid, who was fearful of some of their initiatives. As Duff's biographer Finola Kennedy has noted, today 'it may be difficult to grasp that in the 1920s and 1930s the Legion represented a revolutionary concept of stunning dimensions, anticipating as it did the teaching of the Second Vatican Council' (Kennedy 2011).

Since Vatican II, some of the movements that have made their way to Ireland include the Neocatechumenal Way, Charismatic Renewal, *Focolare*, Marriage Encounter, *L'Arche*, Communion and Liberation, the Emmanuel Movement, and, more recently, Youth 2000. At Pentecost 1998, representatives from these and other movements came together in Rome at the invitation of Pope John Paul II. Addressing them, the Pope declared:

Today a new stage is unfolding before you: that of ecclesial maturity. This does not mean that all problems have been solved. Rather, it is a challenge. A road to take. The Church expects from you the "mature" fruits of communion and commitment.

In our world, often dominated by a secularized culture which encourages and promotes models of life without God, the faith of many is sorely tested, and is frequently stifled and dies. Thus we see an urgent need for powerful proclamation and solid, in-depth Christian formation. There is so much need today for mature Christian personalities, conscious of their baptismal identity, of their vocation and mission in the Church and in the world! There is great need for living Christian communities! And here are the movements and the new ecclesial communities: they are the response, given by the Holy Spirit, to this critical challenge at the end of the millennium. You are this providential response. (Pope John Paul II 1998)

People are attracted to these movements by their strong sense of community and the charism of their founders. One member of *Focolare* describes her experience as one rooted in unity and the quest for peace:

> When I started university, having grown up in Belfast during the Troubles, I wanted to find a way to work for peace. Through Focolare I rediscovered the gospel and became involed with other young people in trying to put it into practice in everyday life. I became more and more fascinated by the ideal of universal brotherhood, and through my membership of this movement I meet many people from all over the world living for the same aim, choosing a lifestyle based on gospel values and dialogue. (Majury 2015)

As the 21st century dawned, these movements may have come of age. They continue to be, for the most part, centred on cities and larger urban areas and may not have impacted significantly on parishes where most of the life of the Church still exists in Ireland. However, their particular charisms and zeal could have a contribution to make to the mission of the Catholic Church in Ireland in the 21st century. An example of this too is the growth (albeit since the Jubilee Year 2000) of the Divine Mercy devotion in Ireland and elsewhere. Many parishes too have embraced this.

## 7. Conclusions

Though not separated from the liturgy but rather, anticipating it and flowing from it, popular piety is an expression of the memory of the people, both historically and in terms of the current needs and affections. It brings people into dialogue with the transcendental. Popular piety is born out of the Christian faith of people and is nourished and grown by both their culture and experience as well as the theology of the church. As we have seen, devotional and popular piety in Ireland was largely personal; it was about removing guilt and attaining salvation for the soul. The mandate of service to others, especially the poor, was not as prevalent as it might have been. Faith was personal, compliant, and overly legalistic. This, in turn, has implications, as it separates spirituality from theology; it also means that active participation cannot attain its true meaning and potential for enhancing the living faith.

Writing in the *Nenagh Guardian* a week after the introduction of the vernacular into the Mass in 1965, Fr Kieran O'Gorman, the diocesan director of music in Killaloe, said that some would be

> inclined to regard all these changes as mere gimmicks and that is what they will become if they are carried out only because they have been imposed by authority'. Offering an assessment of authentic reception, he added '[i]t is only if they are seen to be right and fitting, are accepted with inner conviction, that they are important for a full offering of the sacrifice of the Mass, and are out into practice with a living faith and earnestness, that they will be fruitful. (Nenagh Guardian, 13 March 1965)

Almost twenty years later, one of the great theological minds of the twentieth century, Hans Urs von Balthasar insisted 'on the inseparability between theology and spirituality, their separation being the worst disaster that ever occurred in the history of the Church' (*L'Observatore Romano*, English ed., 23 July 1984). Paul Murray, an Irish Dominican theologian, posits that theology that is 'rather dry and abstract' can be 'almost unrelated to a path of spirituality' and therefore 'out of contact with living faith experience' (Murray 2010). He argues this separation has also impacted on devotional life too as, devoid from the challenge of 'robust intellectual tradition, many of the more popular devotional practices within the Church have tended to assume exaggerated and sentimental forms'.

The hierarchical nature of the reception of the council in Ireland did not allow for the level of active participation that the council demanded because it failed to enable a degree of intellectual engagement that facilitated dialogue between the theology underpinning the council and the living faith experience of Irish Catholics in a way which could integrate

the two. The application of the conciliar vision was driven by a determination on the part of the bishops not to disrupt the faith life of Irish people. This 'top-down' approach, and the sense of obligation and Irish Catholics, coupled with 'timid and rigid conformity' that was inherent in the whole approach to the council, meant that the richness of what Vatican II set forth for the People of God was not attained easily. In Ireland, a country which had great tradition of popular piety and devotion, the evidence shows that there was a chasm between what people prayed and their sense of wider participation in the life of the church. This was made apparent by the period of Vatican II, as liturgical and other changes in terms of the church's self-understanding became a reality in the experiences of the people.

More recently, in 2012, Cardinal Jorge Bergoglio (since Francis 2013) spoke of 'the theology of the people', emphasising that popular piety is the antithesis of widespread secularisation. According to Bergoglio, this theology was popular in Argentina as an alternative to radical liberation theology because it was founded on the culture and devotion of the common people, including their spirituality and sense of justice. Moreover, he states that '[p]opular spirituality is the original way through which the Holy Spirit has led and continues to lead millions', adding that popular piety can be identified as 'the disclosing of the memory of a people'. Therefore, he concludes, 'it is good and necessary that theology cares for popular piety' (*National Catholic Register* 28 April 2015). In the document that set forth the vision of his papacy in 2013, Francis stated that '[p]opular piety enables us to see how the faith, once received, becomes embodied in a culture and is constantly passed on. Once looked down upon, popular piety came to be appreciated once more in the decades following the Council'. Francis sees the role of popular piety as being missionary, adding that it should not be controlled or stifled (Francis 2013). He reinforced this again in 2018 when, addressing the rectors of religious shrines around the world, he reminded them of their duty to provide for people who 'willingly gather to express their faith in simplicity, and according to the various traditions that have been learned since childhood', adding that shrines are irreplaceable 'because they keep popular piety alive, enriching it with a catechetical formation that sustains and reinforces the faith and at the same time nurtures the testimony of charity' (Francis 2018). He went on to refer to popular piety as 'a gem' and also 'the immune system of the Church'.

As the Roman Catholic Church in Ireland moves further into the twenty-first century, a question to be answered is this: 'what is our memory'? What is our memory as regards popular piety and its rich contribution to the liturgical and prayer life of communities and of individuals? In trying to receive the council in a solely liturgical and conformist way, there is a danger that we can throw away the memory of the people without recognising and respecting the great contribution adds to the participation in and the catechesis on that same liturgical life. Theology and spirituality, faith and culture, memory and the experience of God in the here and now of people's lives, together with active participation: all of these in dialogue must be key elements to the pathway ahead for the renewal of the Catholic Church in Ireland.

**Funding:** This research received no external funding.

**Institutional Review Board Statement:** Not applicable.

**Informed Consent Statement:** Not applicable as the study did not concern humans.

**Data Availability Statement:** Archives are accessible by members of the public and scholars.

**Conflicts of Interest:** The author declares no conflict of interest.

## Notes

1. In a report to Rome in 1714, the Bishop of Clogher Hugh McMahon referenced the practice of people giving signals to each other of the time that Mass would be celebrated so that they, though not able to be present in person due to the dangers involved, could kneel and join in spiritually at the appointed time. See (Flanagan 1954).

2. Speaking at the MacGill Summer School, in Glenties, County Donegal, on 24 July 2012, Archbishop Diarmuid Martin of Dublin stated: 'If anything [Vatican II] was a Council which brought us backwards, it brought us back beyond what we had experienced

in our youth and education to a deeper understanding of the faith of the Church, which was rooted in the scriptures themselves and in the constant tradition of the Church'. See, http://www.macgillsummerschool.com/the-catholic-church-in-ireland-turning-the-corner-of-renewal/ (accessed on 2 December 2020).

3    The archives of the Benedictine Abbey at Glenstal hold a wide range of papers concerning the congresses. The papers delivered at Glenstal and a range of commentaries on them were published in three volumes in 1961, 1962 and 1967. See, Dom Placid Murray (ed.), *Studies in Pastoral Liturgy*, 1, (Maynooth: The Furrow Trust, 1961); Vincent Ryan (ed.), *Studies in Pastoral Liturgy*, 2, (Maynooth: The Furrow Trust, 1962); Dom Placid Murray (ed) *Studies in Pastoral Liturgy*, 3, (Maynooth: The Furrow Trust, 1967). I am grateful to Brian O'Shea OSB, Glenstal, for this information.

4    *Irish Press*, 16 Aug 1956, 'Statement by National Synod'. This statement was contained in a message to the Irish people from the National Synod held at Maynooth in 1956, the last such national synod to be held in Ireland.

5    J. McNamee was Bishop of Ardagh and Clonmacnoise (1927–1966) and chairman of the Liturgical Commission of the Irish hierarchy from 1955 until 1963. His interest in liturgy brought him to introduce the dialogue Mass at St Macartan's Cathedral, Monaghan in the early 1920s when he was a priest of Clogher diocese.

6    Lenny to Fergus, 1 February 1961; Fergus to Lenny, 3 February 1961; D'Alton to Fergus, 29 March 1961; Harty to D'Alton, 21 April 1961; D'Alton to Harty, 25 April 1961(Cardinal Ó Fiaich Library and Archives, hereinafter referred to as COFLA), John D'Alton papers, ARCH/12/3/2, Box 4.1).

7    McQuaid, 'Address to Priests Retreat, 13 July 1961, handwritten notes' (Dublin Diocesan Archives, hereinafter referred to as DDA), McQuaid papers, AB8/B/LVII/406).

8    The submissions are published in the *Acta et Documenta Concilio Oecumenico Vaticano II Apparando, Series I, Vol. II, Pars III: Europe, (Hibernia)* (Vatican City, MCMLX), pp 63–109, [hereinafter referred to as *AD* with page references].

9    Submission by Bishop Patrick O'Boyle, 14 October 1963; No. 6.65, Submission of the same date by Quinn, Walsh, Morris, O'Doherty, Fergus, Kyne, Farren, Hanly, Rodgers, Browne and William Brennan (of Toowoomba, Australia). Walsh added, in his own handwriting: '*Canonem Missae mutare non debemus*'—'we ought not change the canon of the Mass' (Archivio Segreto Vaticano (hereinafter referred to as ASV): Concillii Oecumenici Vaticani II, Busta 148, no. 6.19).

10   Submission by Conway, Ahern, Birch, Farren, Fergus, Kyne, Lucey, Morris, Moynihan, Murphy, O'Boyle, O'Doherty, Philbin, Quinn and Rodgers (ASV: Concillii Oecumenici Vaticani II, Busta 149, no. 2.5).

11   ASV: Concillii Oecumenici Vaticani II, Busta 151, No's. 14.43, 15.21, 16.2 to 16.22.

12   Archbishop McQuaid in his instructions dated 19 January 1965 concerning the introduction of the vernacular directed that only Sunday Masses were affected. Latin was to be the only language used at all weekday Masses and those celebrated at side altars on any day. (DDA, McQuaid papers, AB8/B/LVII/522).

13   Conway to all Irish bishops, 2 May 1964, enclosing details of the submissions by the South African and Australian hierarchies to the Holy See (Galway Diocesan Archives (hereinafter referred to as GDA), Browne papers, B/7/B/v/153).

14   The letter enclosing these petitions included a cover note, dated 2 May 1964, from Conway detailing the procedures to be followed and highlighting 'the principle of gradualism, in order to avoid too sudden a change from the present position of the almost exclusive use of Latin, to the new situation in which a greater use of the vernacular is envisaged'. Every decision in relation to the liturgy required a two-thirds majority. (See, GDA: Browne papers, B/7/B/iii/88.)

15   See 'Minutes of General Meeting of the Irish Bishops, 7/8 October 1969' (GDA, Browne papers, B/7/B/iii/96).

16   'Introduction' to *Diocese of Clogher Mass and Prayer Book*, Lent 1977, Clogher Diocesan Archives, Mulligan papers.

17   Mitchell to Browne, 18 February 1964 (GDA, Browne papers, B/12/B/109).

18   Horan to Browne, 8 February 1964 (ibid).

19   Byrne to Browne, 14 February 1964 (ibid). During the priest's recitation of the Nicene Creed, the people recited the text of the Apostle's Creed (a shorter and simpler text) instead.

20   Hynes to Browne, 23 March 1964 (ibid).

21   'Address by Archbishop McQuaid to Priests' Retreat, 3 July 1968' (DDA, McQuaid papers, AB8/B/LVII/604)**.**

22   Browne to Conway 3 January 1970 (GDA: Browne papers, B/11/D/243); Conway to Browne, 7 January 1970 (GDA, Browne papers, B/7/B/v/158).

23   'Minutes of meeting held on 17 June 1974' (GDA, Browne papers, B/7/A/vii/31—'Minute Book of the *Consilium Presbyterate* 1966–1976'). Earlier, in 1972, the Council of Priests in Kildare and Leighlin had raised the possibility of male religious assisting with the distribution of holy communion. See Patrick McGoldrick, to Bishop John McCormack, 5 September 1972 (GDA, Browne papers, B/7/B/v/159).

24   Conway to all bishops, 29 March 1976, enclosing extracts from the relevant minutes (GDA, Browne papers, B/11/D/249 (1).

25   (Irish Episcopal Commission for Liturgy 1994).

26   'Annual Reports on the work carried out on the Cathedral 1957–1966. Annual Progress' (GDA, Browne papers, B/6/C/62 (2)).

27   'Address by Cardinal Conway in Clofeacle parish' (COFLA, Conway papers, File 18/8).

28  Based initially in Portarlington, it was moved to Carlow in 1978 and then to Maynooth in 1996 where it is now the National Centre for Liturgy, incorporated into the Pontifical University Maynooth.

29  'Corpus Christi Procession, Cork, 31 May 1970', Conway papers, COFLA: Press Releases 1970.

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
