# Peer review of "‘Scrupulous and Timid Conformism’: Ireland and the Reception of the Liturgical Changes of Vatican II"

_religions, doi:10.3390/rel12070545_

Round 1

Reviewer 1 Report

this is an excellent essay with a clear argument, effective establishment of context, and smart example of reception history.  it will be of great use to those interested in the out working of Vatican reforms.  -removed for peer review-.

Author Response

Thank you for your review.

Reviewer 2 Report

Although I am not a theologian myself, I was pleased to read the article because Hungary is preparing for the World Eucharistic Congress and the article is a very thorough analysis of the impact of the post-Vatican II liturgical reform in Ireland. 

-removed for peer-review- perhaps a shorter version could be considered. In some places and phenomena the author gives too detailed explanations, for example: lines 28-31, 407-411, 645-651, 754-757, 799-803.

I have only one question, related to the very beginning of the article. In lines 92-93 the author writes:

„Therefore, prayer in the presence of the Blassed Sacrament was seen as being more spiritually satisfying than receiving the Eucharist itself.” Is this statement based solely on Hartigan's article? I do not consider Hartigan's survey, based on a specific source, to be sufficient. Perhaps some additional evidence could be added here. 

I would also suggest that in the section on architectural changes, you include pictures to illustrate your paper.

Author Response

Thank you for your review. 

I have taken on board your suggestions in virtually all of the points. The one where I left the script the same is in relation to Armagh Cathedral and the more recent renovations, which, I believe, represent a better honouring of our heritage. 

Your point in respect of Hartigan is a good one and I have drawn from some of my own research work more locally to give further examples in support of the point I am making. 

Wishing you every blessing now and for the Eucharistic Congress in Budapest. 

Gary Carville 

Reviewer 3 Report

This article is very academically sound, supported by references, and coherent in achieving its objectives. The writing style is very engaging and the use of examples spanning the history was particularly helpful in understanding the author's intent.

Author Response

Thank you for your review. You are very kind.

Gary